# Attraction of Egg Parasitoids *Trissolcus mitsukurii* and *Trissolcus japonicus* to the chemical cues of *Halyomorpha halys* and *Nezara viridula*

**DOI:** 10.3390/insects13050439

**Published:** 2022-05-06

**Authors:** Marica Scala, Jalal Melhem Fouani, Livia Zapponi, Valerio Mazzoni, Karen Elizabeth Wells, Antonio Biondi, Nuray Baser, Vincenzo Verrastro, Gianfranco Anfora

**Affiliations:** 1Center Agriculture Food Environment, University of Trento, Via E. Mach 1, 38098 San Michele all’Adige, Italy; jalal.fouani@unitn.it (J.M.F.); gianfranco.anfora@unitn.it (G.A.); 2Research and Innovation Center, Fondazione Edmund Mach, Via E. Mach 1, 38098 San Michele all’Adige, Italy; livia.zapponi@fmach.it (L.Z.); valerio.mazzoni@fmach.it (V.M.); karen.wells@fmach.it (K.E.W.); 3Department of Agriculture, Food and Environment, University of Catania, Via Santa Sofia 100, 95123 Catania, Italy; antonio.biondi@unict.it (A.B.); 4Department of Mediterranean Organic Agriculture, Mediterranean Agronomic Institute of Bari (CIHEAM Bari), Via Ceglie 9, 70010 Valenzano, Italy; baser@iamb.it (N.B.); verrastro@iamb.it (V.V.)

**Keywords:** *Trissolcus mitsukurii*, *Halyomorpha halys*, behavioural trials, biological control, risk assessment

## Abstract

**Simple Summary:**

When an alien species reaches a new environment, the natural enemies present in that habitat might fail to regulate its population as they might not be host-adapted. Hence, the best solution might be the use of alien biological control agents that are co-evolved with the exotic pest in question. This is the case of *Halyomorpha halys*, which is native to Asia and has recently invaded Europe and the Americas. *Trissolcus japonicus* and *Trissolcus mitsukurii* are among its main parasitoids. Adventive populations of the latter were found in Northern Italy, suggesting its employment within augmentative biological control. Homologous programs with *T. japonicus* are already ongoing in Italy. This procedure implies releasing the parasitoid to increase its population and spread to new areas invaded by *H. halys*. However, a fundamental aspect that must be investigated is the risk-assessment beforehand, i.e., the systematic process of identifying the hazard associated with such a procedure. In this context, the preference of *T. japonicus* and *T. mitsukurii* between two stinkbugs was evaluated in this study. We found that *T. japonicus* preferred the naturally released traces of *H. halys* while *T. mitsukurii* exhibited a higher preference for the natural and synthetic chemical cues of *N. viridula*.

**Abstract:**

*Trissolcus mitsukurii* and *Trissolcus japonicus* are two Asian egg parasitoids associated with different pentatomids such as *Halyomorpha halys*. Adventive populations of *T. mitsukurii* were found in Northern Italy, suggesting its employment as a biological control agent (BCA) against *H. halys*. Nevertheless, to reduce the latter’s population, *T. japonicus* was released in Italy. Releasing an exotic parasitoid requires investigating the interaction between the BCA and the environment to avoid negative impacts on the entomofauna of the new habitat. *Trissolcus mitsukurii* is mainly associated with *Nezara viridula* in its native area. Therefore, we investigated and compared the ability of female *T. mitsukurii* and *T. japonicus* to distinguish between naturally released cues of *H. halys* and *N. viridula.* A single parasitoid was exposed to contact kairomones of both pests to evaluate its modifications in orthokinetic and locomotory behaviour. The behaviour of female *T. mitsukurii* was also tested on synthetic compounds simulating the cues of the two pentatomids. When naturally released cues were used, *T. japonicus* preferred the traces of *H. halys,* while *T. mitsukurii* preferred *N. viridula*’s cues. Moreover, the attraction *of T. mitsukurii* to *N. viridula*’s cues was confirmed with synthetic cues. Additional studies are needed to judge if this parasitoid can be used as a BCA.

## 1. Introduction

*Trissolcus mitsukurii* and *Trissolcus japonicus* Ashmead (Hymenoptera: Scelionidae) are micro-Hymenopteran egg parasitoids. Both are thought to be native to eastern Asia, including Japan and China. *Trissolcus mitsukurii* is mainly associated with *Nezara viridula* (L.) (Hemiptera: Pentatomidae), and for this reason it was allowed to be released in Brazil and Australia for the control of *N. viridula*, a stinkbug pest [1,2,3]. *Trissolcus mitsukurii* has been reported to attack several pentatomid species, among which it presents a considerable rate of parasitism on eggs of the brown marmorated stink bug (BMSB), *Halyomorpha halys* Stål (Hemiptera: Pentatomidae) [4,5,6,7]. It is thought that the range of *T. mitsukurii* could be spreading following the global spread of BMSB [8]. BMSB is an invasive pest native to Eastern Asia that is now widespread in various European regions, including Northern Italy [9,10]. It is a highly polyphagous pest, as it can attack many host plants depending on the season. This results in substantial damages and economic losses, owing to its feeding with its piercing sucking mouthparts [11,12]. *Trissolcus japonicus* (known as the samurai wasp) is also an Asian egg parasitoid of BMSB. In China, it has been able to reach a 90% parasitism rate [7,13]. Both *T. japonicus* and *T. mitsukurii* have been found in multiple regions of Northern Italy and have been shown to have a greater impact in controlling BMSB when compared with other indigenous species of egg parasitoids [14,15,16]. With the attempt to reduce the population of this harmful pest, *Trissolcus japonicus* has been released strategically in Italy. Given the considerable presence of *T. mitsukurii* individuals in northern Italy [16], this parasitoid could potentially be used as a biological control agent (BCA) in the framework of an integrated pest management (IPM) of BMSB. Handling an exotic parasitoid with the intention of augmenting its adventive populations should of course be carefully planned by studying its interactions with native potential host populations and highlighting its preferences [17]. Therefore, to avoid any negative impact on the native fauna, risk assessment is necessary for the potential use of *T. mitsukurii* as a BCA.

Whenever parasitic Hymenoptera perceive signals of the host presence, a series of behavioural steps are initiated. The successful recognition of a potential host depends on these. Such steps account for host habitat location, host location and host selection [18]. Among the factors that lead to the host selection, infochemicals play the most important role [19]. Infochemicals are chemical cues that concern the interaction between different organisms [20,21]. Perceived by the olfactory system at a long distance, when in the form of volatile substances, and by gustatory receptors at a short distance, when in the form of contact substances, infochemicals can allow the detection of an infested area and the appropriate host stage [18]. Kairomones are a type of infochemical that mediate the relationship between organisms belonging to different species, favoring the perceiver [19,21]. Parasitic wasps are known to benefit from them for foraging purposes, in that they induce a motivated search behaviour when received [18,21]. Moreover, parasitoids can use indirect host-related cues, vicariously associated with the host, in order to get into the vicinity of the suitable host stage [21,22]. The strategy of using cues from an unsuitable host stage to reach the right stage is called infochemical detour [19].

When walking on a surface, insects leave traces such as hydrocarbon compounds that remain as footprints, constituting a signal for other species including egg parasitoids [22]. Footprints appear to derive from the insect cuticle and are lipophilic compounds of low volatility secreted by the tarsal pads when in contact with the plant surface [23,24,25]. Diglycerides and triglycerides of high molecular weight, along with long-chain alcohols and fatty acids, were found in *N. viridula* footprints [26]. The epicuticular waxes of the plant can retain chemical footprints by virtue of their lipophilic properties, allowing the traces to be detected by egg parasitoids as kairomones [19,25]. In the literature, there are different studies about the use of contact kairomones by scelionid parasitoids [27,28,29]. When perceiving host footprints, parasitoids can adopt a systematic or random search. The latter is usually used when the cues derive from a host stage indirectly associated with the suitable ones [18,30]. Colazza et al. (2009) [30] reported that platygastrid egg parasitoids, in response to adult pentatomid footprints, increase the host searching behaviour moving around the contaminated area and return to the chemical cues after losing track of them. Egg parasitoids such as *Trissolcus basalis* and *T. brochymenae* displayed decreased walking speeds when exposed to a substrate contaminated by the adult host [27,28]. The preference for the coevolved host was shown in *Trissolcus* wasps [31], suggesting their capacity to discriminate among different host species with the purpose of choosing the most suitable ones [29,32].

In this framework, the aim of the present work was to assess and compare the discrimination capability of *T. japonicus* and *T. mitsukurii* between the chemical cues of *N. viridula* and *H. halys*. Considering that several previous studies on the searching behaviour of *T. japonicus* proved to have a positive response for *BMSB* footprints, here, we tested its foraging behaviour applying the same approach [33,34]. In the trial with *T. mitsukurii*, we used both chemical cues naturally released by adult female stink bugs, as well as specific ratios of synthetic compounds that simulate the main volatile cues addressed to the two stink bugs. The outcome of this study could contribute to a better understanding of the recognition mechanisms to the different pentatomid hosts by *T. mitsukurii,* allowing us to predict the potential effects of its use as a BCA through augmentation strategies.

## 2. Materials and Methods

### 2.1. Parasitoids

*Trissolcus mitsukurii* and *T. japonicus* were collected in Trento province in summer 2019 [15], and colonies were established in the quarantine facility at Fondazione Edmund Mach (Italy). The parasitoids were reared in plastic vials (50 mL), at 25 ± 1 °C, 60 ± 5% RH and a 16:8 day:night photoperiod. Pure honey droplets were supplied and replaced twice a week. Stored (at −80 °C, up to 6 months) *BMSB* egg masses were exposed to female specimens for progeny production.

### 2.2. Hosts

The BMSB colony was established from overwintering adults collected in Trento province in October 2019 using live traps [35]. The insects were kept in mesh rearing cages (BugDorm ^®^, Taichung, Taiwan, 30 × 30 × 30 cm) at 25 ± 1 °C, 60 ± 5% RH and a 16:8 day:night photoperiod, provided with French beans (*Phaseolus vulgaris* L.), tomatoes (*Solanum lycopersicum* L.), carrots (*Daucus carota* L.) and raw peanuts (*Arachis hypogaea* L.). Food and water were replaced twice weekly. The cages were inspected daily to collect fresh egg masses that were used for the rearing of the parasitoids.

*Nezara viridula* specimens for experiments were collected in Trento province in June 2020 and reared following the same process used for BMSB.

### 2.3. Foraging Behaviour on Footprints

The foraging behaviour of *T. mitsukurii* and *T. japonicus* on BMSB and *N. viridula* footprints was assessed using cellulose filter papers (6 cm ø, VWR^®^ n.401) contaminated by the cues laid by an adult female of each of the two pentatomid species. The abovementioned filter papers were obtained by forcing a mated female pentatomid to walk on the filter papers inside a plastic Petri dish (6 cm ø) for 30 min. Uncontaminated filter papers were used as control. Filter papers with frass deposits were discarded. The experimental arena consisted of a glass Petri dish (10 cm ø) with the filter paper, placed on a light panel (Medalight LP-1218) to attract the parasitoid to the lower surface and thus to increment the color contrast and to facilitate the behavioural obeservations. The behavioural responses were recorded using a video camera (CANON EOS 80D). Each filter paper was employed for five observed individual parasitoids, i.e., for five replicates. The no-choice trials were performed using a naïve mated individual female (N = 30) of *T. mitsukurii* or *T. japonicus* (3–5 days old) and placed inside the arena. The observations lasted for a maximum of 10 min and were interrupted after one cumulative minute spent outside the filter paper (behaviour that indicates no interest in the filter paper). The recorded videos were analyzed with Ethovision ^®^ XT 9 software. The variables considered for investigating the *T. mitsukurii* preference towards the two pentatomids were angular velocity mean (deg/s), in-zone cumulative duration (i.e., total residence time on filter paper in seconds), mean velocity (m/h) and distance moved (cm). The trials were carried out in a dark climatic chamber, lit only by the light panel, at 26 ± 1 °C and 70 ± 1% RH from 10.00 to 12.00.

### 2.4. Foraging Behaviour on Synthetic Compounds

The exposure of *T. mitsukurii* to the synthetic chemical cues of BMSB and *N. viridula* was carried out considering the naturally released compounds of the two pentatomids [33,36], using their main components. These components were tridecane, (*E*)-2-decenal and (*E*)-2-hexenal (Sigma-Aldrich, Sent Luis, MO, USA). According to Malek et al. (2021) [33], the 4:1 ratio of tridecane to (*E*)-2-decenal (1.6: 0.4 nl/mL) has a significant positive effect on the foraging behaviour of *T. japonicus*. In this study, we used the same ratio for *T. mitsukurii* in the case of BMSB. As for *N. viridula,* we used a 3:1 ratio for the two main components of its footprints, tridecane and (*E*)-2-hexenal (1.5: 0.5 nl/mL), as indicated by the study of Aldrich et al. (1978) [36]. Pure compounds were prepared using aliquots containing 0.2 nL of each compound in 100 µL of the solvent dichloromethane (Sigma-Aldrich Chemie, Steinheim, Germany). For the control, 100 µL aliquots of the solvent dichloromethane were deployed. The above-mentioned compounds were applied on sterile filter papers (6 cm ø) and left to dry for 2 min. The experimental design remained the same as in footprint trials.

### 2.5. Statistical Analysis

The differences among behavioural parameter groups were analyzed using Wilks’ Lambda type non-parametric interference for multivariate data [37]. All statistical analyses were performed in R (https://www.R-project.org/, accessed on 16 June 2021) through the application of *nonpartest* and *ssnonpartest* functions in the “npmv” package, which provided post hoc analysis as well [38].

## 3. Results

### 3.1. Foraging Behaviour on Footprints

An interested parasitoid, when triggered by chemical cues released by a suitable host, demonstrates higher angular velocity, slower walking velocity, greater residence time and longer covered distances [23,39].

For *T. japonicus*, the statistical analysis highlighted significant differences among groups (Wilks Lambda = 27.304, df = 8, 168, *p* < 0.001) (Figure 1). Analyzing the foraging behaviour of *T. japonicus* females, a preference for the substrate contaminated by the BMSB female compared to *N. viridula* and the control emerged. The time spent on the BMSB-contaminated surfaces was higher than in *N. viridula* and the control, in that order. The same trend of preference occurred for the mean velocity, where the lowest values were associated with BMSB, succeeded by *N. viridula* and the control. The distance moved was the highest for BMSB. Similar to the distance moved, the greater recorded value was for BMSB, followed by *N. viridula* and the control. The values featured in Table 1 represent the probability that a value obtained from one treatment sample is higher than a randomly chosen value of the other treatment samples.

In the trial with *T. mitsukurii*, a significant difference between groups was observed (Wilks’ Lambda = 8.490, df = 8, 168, *p* < 0.001) (Figure 2). The tested parasitoids showed a clear preference toward *N. viridula* footprints, in comparison to those released by BMSB and the control. The time spent on filter paper was higher for *N. viridula*, followed by BMSB and control, respectively. Greater values for mean velocity were recorded for the control. As for the time spent on filter paper, higher values were observed for *N. viridula* in the distance moved, followed by BMSB and the control. The same trend was recorded for the angular velocity as well (Table 1).

### 3.2. Foraging Behaviour on Synthetic Compounds

Regarding the trials with the use of synthetic compounds, there was a significant difference between the three treatments (Wilks’ Lambda = 3.397, df = 8, 168, *p* = 0.001) (Figure 3). In particular, *T. mitsukurii* female parasitoids preferred *N. viridula* synthetic chemical cues. The data collected revealed no significant preference for the BMSB mixture compared to the other two treatments. Nevertheless, the parasitoids showed low interest for the uncontaminated filter papers. Higher values in terms of time spent on the filter paper and distance moved were observed for *N. viridula* compared to the control and BMSB. The mean velocity was lower for BMSB and *N. viridula,* respectively. Greater values were displayed in the angular velocity for BMSB, *N. viridula* and control, in that order. The values featured in Table 1 represent the probability that a value obtained from one treatment sample is higher than a randomly chosen value of the other treatment samples.

## 4. Discussion

Egg parasitoids have adapted to employ infochemical detours from herbivore adults, with the aim of detecting the suitable host stage [19]. Chemosensors are used to recognize odors produced in minute concentrations by hosts, while the antennae and the brain perform a filter function, reducing the complexity of the stimuli received [40,41]. This elaborate mechanism provides details about the presence of the host and, consequently, parasitism success. Kairomones act as signals, inducing in *Trissolcus* wasps some specific habits consisting of arrestment behaviour characterized by slow-going speed and small distance displacement [28,33]. Studies that concern the interaction between a parasitoid and host trails, involving chemical ecology, should be considered as an essential integration to conventional host no-choice experiments (physiological host range) to understand the actual host range under conditions that mimic as closely as possible natural ones and incorporate part of the environmental complexity. In this regard, this work provides elements about the ecological host range of *T. mitsukurii*, investigating its principal hosts as per the literature.

In the present study, for both parasitoids *T. mitsukurii* and *T. japonicus*, an alteration in their orthokinetic and locomotory behaviour was noticed, which was elicited by chemical traces released by the two pentatomids. The preference towards their respective primary host was demonstrated. Indeed, *T. mitsukurii* manifested a deceleration when in contact with *N. viridula* traces, probably to better perceive the trail marked out by the host. In contrast, *T. japonicus* revealed a preference for BMSB cues, with a higher residence time and lower velocity. The use of a synthetic mixture that mimicked *N. viridula* cues induced an arrestment response in *T. mitsukurii*, exhibiting an inclination in the choice of this pest.

When investigating singular parameters, the angular velocity revealed the most heterogeneous response in different trials. This parameter indicates the change in direction, meaning the more the parasitoid returns to the trails, the higher its value will be. Various studies demonstrated a discrepancy in the behavioural response connected to this parameter. Indeed, for *T. basalis*, when it comes to angular velocity, no significant difference was found in patches contaminated by different bugs and the control, despite the residence time being significantly higher in patches with *N. viridula* kairomones [29]. *Trissolcus brochymenae* exhibited no difference between chemical traces of different host stages and the control in the angular velocity as well [27]. On the contrary, Boyle et al. (2020) [34] illustrated a greater angular velocity in *T. japonicus* on leaves contaminated by its main host BMSB, even if no difference existed among treatments with the use of soybean leaf as substrate. In this study, the remaining parameters agreed with similar previous tests, showing a certain preference for one species over another. In previous studies, *T. japonicus* experienced a stronger arrestment behaviour to BMSB footprints compared to those of *Podisus maculiventris* (Hemiptera: Pentatomidae), a suboptimal host species, showing higher residence time and covering greater distance in response to BMSB female kairomones [33]. The same pattern occurred in the residence time with the use of a contaminated leaf as a substrate, though no significant differences were observed between the two groups in the linear walking speed [34]. This proves that the contact with the treated surface could affect locomotory paths in different manners, denoting an interest for the attractive substances produced by stink bugs in relation to the control substrate. Conti et al. (2003) [27] noticed the ability of *T. brochymenae* to discriminate among various stages, different sexes and physiological conditions of its host *Murgantia histrionica* (Hemiptera: Pentatomidae). Indeed, the female parasitoids preferred to spend more time on the patches contaminated by gravid or virgin pentatomid females, whereas there was no difference in the residence time between patches with virgin pentatomid male kairomones and the control ones. The host’s physiological condition influenced the arrestment behaviour of *T. basalis* when it encountered chemical traces of *N. viridula*, revealing an arrestment response and a reduced speed for pre-ovipositional females [28]. The use of arenas contaminated by the co-evolved *N. viridula* induced *T. basalis* to spend higher residence time compared to the arenas containing the trails of a non-coevolved host [29]. The locomotory path was affected as well, with a lower linear speed not only in substrate contaminated by *N. viridula* but also by the non-coevolved *M. histrionica* and *Eurydema ventralis* (Hemiptera: Pentatomidae) [29]. Furthermore, the preference for the co-evolved host was observed in *T. brochymenae*, highlighted by the highest residence time in areas contaminated by *M. histrionica*. Nevertheless, *Trissolcus simoni* (Hymenoptera: Scelionidae) reacted to the stimuli of different hosts with similar intensity to that of the co-evolved *E. ventralis* [32]. Thus, it seems clear that *Trissolcus* species have a great discriminatory capability that allows them to select the trails released by the hosts’ stage correlated to the suitable oviposition substrate. This strategy is related to successful reproduction to ensure the generation of a new progeny. Moreover, this ability enables them to discern between co-evolved and non-coevolved hosts.

In the present study, *T. japonicus* revealed a preference for the principal host, BMSB, which is in accordance with literature [33,34]. In contrast, *T. mitsukurii* reacted differently, demonstrating a preference for *N. viridula* kairomones. This detail could presume the existence of some sort of co-evolutionary or adaptation process. A recent survey on the olfactory response of *T. mitsukurii* confirms this inclination for *N. viridula* [42]. Indeed, the parasitoid manifested an interest in the arm containing a plant contaminated by *N. viridula*, which was previously exposed to individuals for feeding and oviposition [42].

Examples of *N. viridula*/*T. mitsukurii* association are present in the literature [1,3], confirming the parasitoid tendency to perform a motivated searching behaviour when entering in contact with the cues of this phytophagous species. However, a recent study on the physiological host range of *T. mitsukurii* surprisingly highlighted the low suitability of *N. viridula* egg masses, with a reduced parasitoid emergence rate and elevated amounts of dead eggs [43]. The introduction of the parasitoid in a new area could have driven a genetic drift, in response to different biotic factors but also in reaction to the evolutionary trap represented by *N. viridula*, which results in a dead end of *T. mitsukurii* progeny. Moreover, the development in BMSB eggs positively influences parasitoids’ fitness, ensuring survival in a foreign zone.

The specimens of *T. mitsukurii* used for our trials were reared on BMSB eggs. Such a condition could have affected the foraging behaviour, causing the wasp to be less assertive in its choice when exposed to the two stimuli [44]. The response to chemical trails may change with the learning process of the parasitoid. The development of a particular host could influence the preadult learning [45,46]; however, contact kairomones are mainly responsible for inbred responses [19]. The degree of host specificity can affect the learning experience as well [44].

## 5. Conclusions

In conclusion, *T. mitsukurii* preferred the contact kairomones produced by *N. viridula*, especially when naturally released when walking. Conversely, BMSB cues elicited a modification in the kinetic reaction with a reduced intensity. The aforementioned interest was confirmed in olfactory investigation [42], while discordant results were highlighted in the physiological host range [43]. Additional analyses are needed to prove this preference. The employment of an exotic BCA to control a pest could imply collateral effects on the entomofauna present in the area of its release. Therefore, the incorporation of chemical ecology studies within the risk assessment might help with the comprehension of the natural mechanisms conducted by the BCA.

## Figures and Tables

**Figure 1 insects-13-00439-f001:**
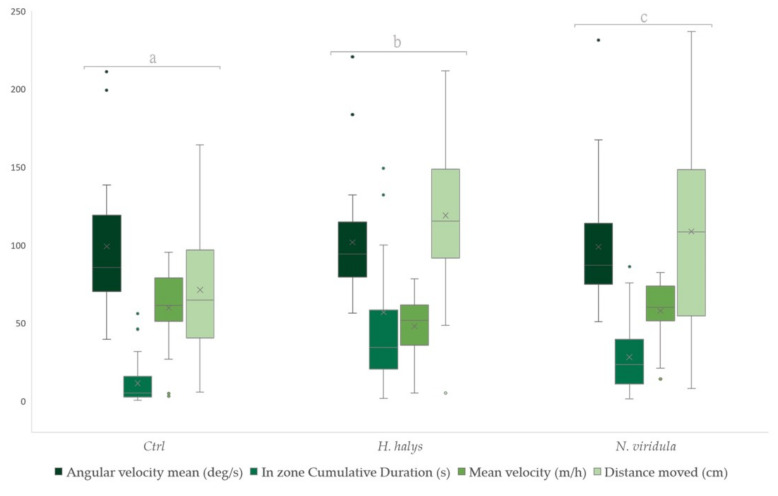
Boxplots of foraging behaviour parameters for females of *T. japonicus* following contact with footprint-contaminated substrate from female adults of *H. halys* and *N. viridula*. Behavioural parameter groups indicated by different letters show significant differences (Wilks’ Lambda type non-parametric interference at *p* < 0.001). Boxplots represent the interquartile range, with a horizontal bar as the median. The bottom whisker and the upper whisker represent the minimum and maximum values, respectively. The “X” symbol represents the mean value. The dots show the outliers.

**Figure 2 insects-13-00439-f002:**
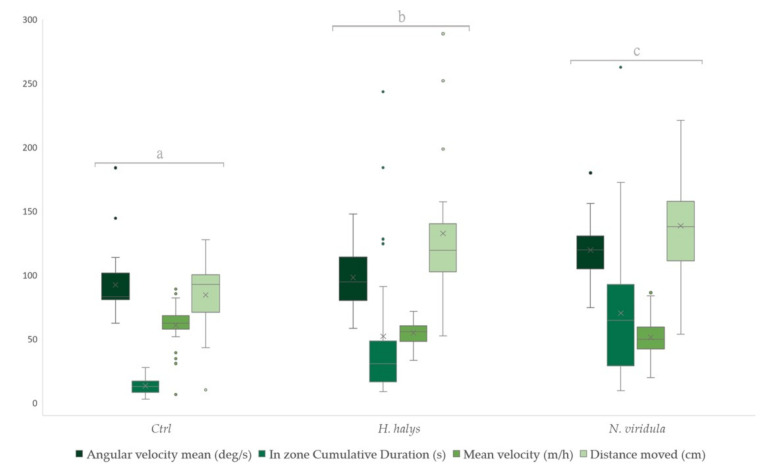
Boxplots of foraging behaviour parameters for females of *T. mitsukurii* following contact with footprint contaminated substrate from female adults of *H. halys* and *N. viridula*. Behavioural parameter groups indicated by different letters show significant differences (Wilks’ Lambda type non-parametric interference at *p* < 0.001). Boxplots represent the interquartile range with a horizontal bar as the median. The bottom whisker and the upper whisker represent the minimum and maximum values, respectively. The “X” symbol represents the mean value. The dots show the outliers.

**Figure 3 insects-13-00439-f003:**
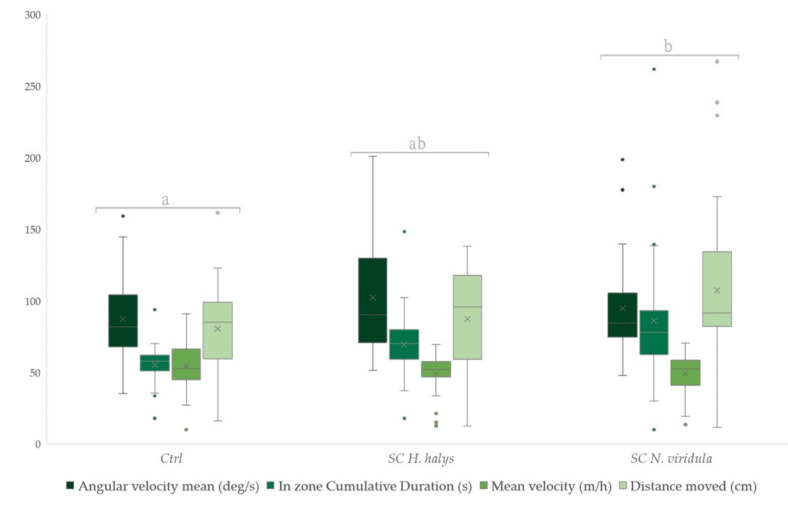
Boxplots of foraging behaviour parameters for females of *T. mitsukurii* following contact with contaminated substrates by synthetic compounds (SC) of *H. halys* and *N. viridula*. Behavioural parameter groups indicated by different letters show significant differences (Wilks’ Lambda type non-parametric interference at *p* < 0.001). Boxplots represent the interquartile range with a horizontal bar as the median. The bottom whisker and the upper whisker represent the minimum and maximum values, respectively. The “X” symbol represents the mean value. The dots show the outliers.

**Table 1 insects-13-00439-t001:** Relative effects that quantify the tendencies observed in the data in terms of probabilities for each species within different treatments (i.e., footprints and synthetic compounds). All reported values were calculated using the npmv package in R program [37].

			Parameters			
			Angular Velocity Mean	In Zone Cumulative Duration	Mean Velocity	Distance Moved
Footprints	*T. japonicus*	Control	0.46852	0.30000	0.56778	0.51556
		*H. halys*	0.53815	0.66815	0.39000	0.73704
		*N. viridula*	0.49333	0.53185	0.54222	0.24741
	*T. mitsukurii*	Control	0.34630	0.24352	0.64074	0.27111
		*H. halys*	0.45815	0.57389	0.47333	0.56407
		*N. viridula*	0.69556	0.68259	0.38593	0.66481
Synthetic compounds	*T. mitsukurii*	Control	0.45185	0.31833	0.54593	0.42704
		SC *H. halys*	0.55111	0.54481	0.47296	0.51074
		SC *N. viridula*	0.49704	0.63685	0.48111	0.56222

## Data Availability

The data presented in this study are available on request from the authors.

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
