# Peer review of "Attraction of Egg Parasitoids Trissolcus mitsukurii and Trissolcus japonicus to the chemical cues of Halyomorpha halys and Nezara viridula"

_insects, 2022, doi:10.3390/insects13050439_

Round 1

Reviewer 1 Report

Review - Insects-1669750 – Scala et al – Attractiveness to chemical cues of stink bugs

Very interesting work of significance to stink bug biocontrol, and broader application to biological control and parasitoid behavior. The study and the results are clearly introduced and described, and I have no significant procedural questions or comments, except as indicated below.

Although the clarity of English is overall adequate for readers to understand the paper, the writers are clearly not native English speakers, as there are numerous instances throughout the text of grammatical errors, imprecisions and awkward translations. The paper will benefit from final editorial assistance by a fluent English speaker to help clear these up.

Lines 147-148     Results from re-using the same filter for 5 replicates could conceivably be compromised by traces left by previous parasitoids and detected by new females. Was this possibility considered?

192        “As for t mitsukurii,…” will suggest to some readers fluent in English that this paragraph refers to results with T japonicus, by comparison with T mitsukurii. This is the opposite of what the authors intend. I recommend changing the leading sentence to simply “In the trials with T mitsukurii…” to eliminate the possible confusion.

223        I think authors mean to use “provision” rather than “prevision” here.

229        Reference 39 does not look like the correct number – should it be 40?

Table 1                 In the figures the grouped behavioral responses were shown as significantly different. It would be helpful to readers for authors to also indicate individual values with significant differences (or not) in the table, such as with asterisks or superscript letters.

Author Response

Comments and Suggestions for Authors

Review - Insects-1669750 – Scala et al – Attractiveness to chemical cues of stink bugs

Very interesting work of significance to stink bug biocontrol, and broader application to biological control and parasitoid behavior. The study and the results are clearly introduced and described, and I have no significant procedural questions or comments, except as indicated below.

Although the clarity of English is overall adequate for readers to understand the paper, the writers are clearly not native English speakers, as there are numerous instances throughout the text of grammatical errors, imprecisions and awkward translations. The paper will benefit from final editorial assistance by a fluent English speaker to help clear these up. The text was revised by a native English speaker.

Lines 147-148     Results from re-using the same filter for 5 replicates could conceivably be compromised by traces left by previous parasitoids and detected by new females. Was this possibility considered? We considered this aspect when setting up the arena for this study. Based on bibliographical research, it seems that there is no impact on parasitoids when filter paper is used for 5 replicates. See (Malek et al., 2021).

192   “As for t mitsukurii,…” will suggest to some readers fluent in English that this paragraph refers to results with T japonicus, by comparison with T mitsukurii. This is the opposite of what the authors intend. I recommend changing the leading sentence to simply “In the trials with T mitsukurii…” to eliminate the possible confusion. Done

223        I think authors mean to use “provision” rather than “prevision” here. We changed the expression with a more suitable one.

229        Reference 39 does not look like the correct number – should it be 40? All references were re-checked and corrected.

Table 1                 In the figures the grouped behavioral responses were shown as significantly different. It would be helpful to readers for authors to also indicate individual values with significant differences (or not) in the table, such as with asterisks or superscript letters. The table shows the “non-parametric relative effects” that are merely calculated probabilities (check the text for the updated description). Therefore, significant differences between single responses are not provided as the used statistical analysis only provides significance for groups.

We thank the reviewer for corrections and constructive suggestions that helped to improve the quality of the manuscript.

Reviewer 2 Report

Review of Insects ms. Scala et al., “Attractiveness of Egg Parasitoids Trissolcus mitsukurii and Trissolcus japonicus to the chemical cues of Halyomorpha halys and Nezara viridula

This manuscript clearly reflects some interesting and apparently sound research to the readers of the journal.  However, there are several general issues requiring widespread revision in the manuscript:

  1. The title must be changed to “Attraction...” instead of “Attractiveness...”. As currently written it means the reverse of what it should say.
  2. The error in the title is symptomatic of errors and odd expressions in the English. I have tried to enumerate corrections but there are many present, and this needs thorough review.
  3. The “Simple Summary” is inadequate. It does not state what the findings were. Both this summary and the Abstract need thorough rewriting (some additional details below).
  4. The introduction and discussion are missing at least the following two key citations: Zhong, Yong-Zhi, et al. "Behavioral responses of the egg parasitoid Trissolcus japonicus to volatiles from adults of its stink bug host, Halyomorpha halys." Journal of Pest Science4 (2017): 1097-1105; Boyle, Sean M., et al. "Parental host species affects behavior and parasitism by the pentatomid egg parasitoid, Trissolcus japonicus (Hymenoptera: Scelionidae)." Biological Control149 (2020): 104324.
  5. Methods are missing some key descriptions, including how the parasitoids were collected, and very importantly what was the method of processing of the video recordings to produce the dependent variables obtained? Also, an explicit statement that the bioassays were “no-choice”.
  6. Figures 1 through 3 purport to show treatment differences in the multivariate outcome, but then the text states that specific outcome variables, e.g. in-zone cumulative duration, differ significantly from one another, yet this is not supported by the statistical test used. This needs to be far better explained and/or include additional statistical tests. Also in the figures, the legibility needs to be improved and the symbol “X” explained in the captions.
  7. It is not clear from Table 1, what the values presented signify, in terms of treatment differences or statistical significance. “Ctrl” should be spelled out, and units for each parameter shown as well.
  8. The discussion is too long. The first paragraph should be deleted; this study does not pertain to risk assessment. How does the parasitoid response reflect an “infochemical detour” if the kairomonal substances may also be on the egg mass itself? The last two discussion paragraphs are too long and too speculative. Conclusions also need to be strengthened to explain what strategy is supported (last sentence).
  9. There are many omissions and errors in the References, for example, (11) journal name improperly abbreviated; (13) no journal name; (18) author name repeated; (20) journal name unclear; (40) reference is not citable as “in review”; (41) is missing volume and page number; (42) is actually a book review (!), not the book itself; (44) journal name improperly abbreviated. Please review ALL references for proper information and consistent format.

Overall these issues require a major revision.  Additional details appear below.

Page 1, line 16, change to “may not be host-adapted”.

L19, are, not “lie”.

L21, biological control

L21, where are “homologous programs” effective?

L23, beforehand, not “before”.

L24, what procedure?

L15-28 The simple summary does not summarize the findings of the study.

L29 & L35, associated with, not “to”.

L32, introduced, not “released”.

L42, state the findings and significance – do not say they “are discussed”.

Introduction – beginning is confusing about T.mitsukurii range – China, Japan?

L57, change to “is able”.

L59, many, not “several”.

L66, “presume” is incorrect word here.

L70, “native potential host populations,” in place of “autochthonous host populations”.

L71, how is this a landmark and how is it “risk assessment” – the two parasitoid species are already present in northern Italy.

L73, omit “BCA” – not used later.

L89 & L344, “deambulation,” while an English word, is better stated as “walkiing”.

L96: “traces” not “it”.

L96-98 is repetitious; delete.

L102 pentatomid, not “pentatomids” – this is the adjective.

L103, them not “it”.

L104, delete commas.

Intro last paragraph needs to be rewritten. Aren’t you comparing (testing for differences in) preferences between the two parasitoid species, with regard to the two hosts?

L121: how were the parasitoids collected?

L150 ff, State explicitly this was a series of no-choice tests (correct?).

L151, 152, filter paper, not “filter”

L156, was the chamber completely dark??

Section 2.4, All positional isomers, e.g. “E” should be italicized, and chemicals not capitalized, e.g. tridecane.

L183, “A slight preference”  -- see point #6 above about lack of clarity in statistical tests.

L197, As, not “like”.

For all figures, see point #6 above.

Table 1, see point #7 above.

L208, 209: State explicitly the evidence for preference – duration?  Which parameter in which direction?

Figure 3 caption: synthetic, not synthetical;. parameter not parameters; indicated not sustained.

Discussion: see point #8 above.

l223, what is “prevision”?

Author Response

Review of Insects ms. Scala et al., “Attractiveness of Egg Parasitoids Trissolcus mitsukurii and Trissolcus japonicus to the chemical cues of Halyomorpha halys and Nezara viridula

This manuscript clearly reflects some interesting and apparently sound research to the readers of the journal.  However, there are several general issues requiring widespread revision in the manuscript:

  1. The title must be changed to “Attraction...” instead of “Attractiveness...”. As currently written it means the reverse of what it should say. Done
  2. The error in the title is symptomatic of errors and odd expressions in the English. I have tried to enumerate corrections but there are many present, and this needs thorough review. The text was revised by a native English speaker.
  3. The “Simple Summary” is inadequate. It does not state what the findings were. Both this summary and the Abstract need thorough rewriting (some additional details below). Done
  4. The introduction and discussion are missing at least the following two key citations: Zhong, Yong-Zhi, et al. "Behavioral responses of the egg parasitoid Trissolcus japonicus to volatiles from adults of its stink bug host, Halyomorpha halys." Journal of Pest Science4 (2017): 1097-1105; Boyle, Sean M., et al. "Parental host species affects behavior and parasitism by the pentatomid egg parasitoid, Trissolcus japonicus (Hymenoptera: Scelionidae)." Biological Control149 (2020): 104324. The citations were integrated into the text.
  5. Methods are missing some key descriptions, including how the parasitoids were collected, and very importantly what was the method of processing of the video recordings to produce the dependent variables obtained? Also, an explicit statement that the bioassays were “no-choice”. We added the missing information.
  6. Figures 1 through 3 purport to show treatment differences in the multivariate outcome, but then the text states that specific outcome variables, e.g. in-zone cumulative duration, differ significantly from one another, yet this is not supported by the statistical test used. This needs to be far better explained and/or include additional statistical tests. Also in the figures, the legibility needs to be improved and the symbol “X” explained in the captions. We never suggested that specific variables are significantly different for each treatment, but rather a significance between groups. The letter “X” was explained accordingly.
  7. It is not clear from Table 1, what the values presented signify, in terms of treatment differences or statistical significance. “Ctrl” should be spelled out, and units for each parameter shown as well. The text was updated for better clarification on the meaning of the values featured in the table. The “Ctrl” was spelled out. We decided to not specify the units for each parameter since these numbers are merely predicted probabilities to support the explanation of the graphs and not the real values. In particular, this aspect is explained in L 188-190.
  8. The discussion is too long. The first paragraph should be deleted; this study does not pertain to risk assessment. How does the parasitoid response reflect an “infochemical detour” if the kairomonal substances may also be on the egg mass itself? The last two discussion paragraphs are too long and too speculative. Conclusions also need to be strengthened to explain what strategy is supported (last sentence). The discussion has been reduced according to your comments. We studied the behavioural responses of trissolcus parastoids in terms of chemical ecology, a branch of science that we retain important when conducting a risk-assessment. The “infochemical detour” is a strategy adopted by the egg parasitoids in order to arrive to the egg masses, fallowing the trails of the adult stage. In this experiment the host trails of mated adult females were used. The conclusion was revised accordingly.
  9. There are many omissions and errors in the References, for example, (11) journal name improperly abbreviated; (13) no journal name; (18) author name repeated; (20) journal name unclear; (40) reference is not citable as “in review”; (41) is missing volume and page number; (42) is actually a book review (!), not the book itself; (44) journal name improperly abbreviated. Please review ALL references for proper information and consistent format. References were revised and edited.

Overall these issues require a major revision.  Additional details appear below.

Page 1, line 16, change to “may not be host-adapted”. Done

L19, are, not “lie”. Done

L21, biological control Done

L21, where are “homologous programs” effective? Done

L23, beforehand, not “before”. Done

L24, what procedure? Refer to L 20.

L15-28 The simple summary does not summarize the findings of the study. Done

L29 & L35, associated with, not “to”. Done

L32, introduced, not “released”. Since the population of T. japonicus was already present, we are not sure that “introduced” is the right word.

L42, state the findings and significance – do not say they “are discussed”. Done

Introduction – beginning is confusing about T.mitsukurii range – China, Japan? Modifications were done to clarify this point.

L57, change to “is able”. Done

L59, many, not “several”. Done

L66, “presume” is incorrect word here. We modified the sentence.

L70, “native potential host populations,” in place of “autochthonous host populations”. Done

L71, how is this a landmark and how is it “risk assessment” – the two parasitoid species are already present in northern Italy. When a BCA is released intentionally with the intent to increase its population, the natural balance is manipulated in favor of the BCA. This is one of the reasons why the risk assessment must be carried out even if the parasitoids are already present in the environment.

L73, omit “BCA” – not used later. It was used in L 31, 33, 42, 66, 71, 115, 318, 321.

L89 & L344, “deambulation,” while an English word, is better stated as “walkiing”. Done

L96: “traces” not “it”. Done

L96-98 is repetitious; delete. Done

L102 pentatomid, not “pentatomids” – this is the adjective. Done

L103, them not “it”. Done

L104, delete commas. Done

Intro last paragraph needs to be rewritten. Aren’t you comparing (testing for differences in) preferences between the two parasitoid species, with regard to the two hosts? The paragraph was reformulated.

L121: how were the parasitoids collected? Done

L150 ff, State explicitly this was a series of no-choice tests (correct?). Done

L151, 152, filter paper, not “filter” Done

L156, was the chamber completely dark?? Yes. We changed the text in order to explain better the experimental set up.

Section 2.4, All positional isomers, e.g. “E” should be italicized, and chemicals not capitalized, e.g. tridecane. Done

L183, “A slight preference”  -- see point #6 above about lack of clarity in statistical tests. Sentence removed.

L197, As, not “like”. Done

For all figures, see point #6 above. Done

Table 1, see point #7 above. The text was updated for better clarification on the meaning of the values featured in the table.

L208, 209: State explicitly the evidence for preference – duration?  Which parameter in which direction? We added a sentence that explains the behaviour of an interested parasitoid (L 177- 179).

Figure 3 caption: synthetic, not synthetical;. parameter not parameters; indicated not sustained. Done

Discussion: see point #8 above. Done

l223, what is “prevision”? We used another expression.

We thank the reviewer for corrections and constructive suggestions that helped to improve the quality of the manuscript.

Round 2

Reviewer 2 Report

Authors have made the appropriate changes in response to both reviews. Thank you for the thorough revisions.